# Femtosecond Laser-Assisted Cataract Surgery versus Conventional Phacoemulsification Surgery: Clinical Outcomes with EDOF IOLs

**DOI:** 10.3390/jpm13030400

**Published:** 2023-02-24

**Authors:** Pingjun Chang, Fan Zhang, Hongzhe Li, Zhuohan Liu, Siyan Li, Shuyi Qian, Yune Zhao

**Affiliations:** 1Department of Ophthalmology, Eye Hospital and School of Ophthalmology and Optometry, Wenzhou Medical University, Wenzhou 325027, China; 2Key Laboratory of Vision Science, Ministry of Health, Wenzhou 325027, China

**Keywords:** femtosecond, phacoemulsification, EDOF IOLs, cataract

## Abstract

In this study, we evaluate and compare the outcomes of conventional phacoemulsification cataract surgery (CPS) and femtosecond laser-assisted cataract surgery (FLACS) with the implantation of an extended depth of field (EDOF) intraocular lens (IOL). A prospective, consecutive cohort study was conducted. Patients were given the option to choose FLACS or CPS and were implanted with an EDOF IOL. Refractive data, visual acuity data, ocular aberration measured with a wavefront aberrometer, and optical quality measured with an optical quality analysis system II were collected at one month postoperatively. A total of 92 eyes of 64 patients were enrolled in this study; 35 eyes of 26 patients were treated with FLACS, whereas 57 eyes of 38 patients were treated with CPS. Uncorrected visual acuity at far, intermediate, and near distance and best-spectacle-corrected visual acuity were not statistically significantly different between the groups (all *p* > 0.05), nor were the mean cylinder and mean spherical equivalent refraction (both *p* > 0.05). The FLACS group had a lower ocular trefoil than the CPS group (*p* = 0.033), and there was no significant difference between the two groups considering other aberration parameters, whether ocular or internal (all *p* > 0.05). Optical-quality-related parameters showed also no significant difference between the two groups (all *p* > 0.05). In conclusion, there was no significant difference between FLACS and CPS with implantation of EDOF IOLs in postoperative ocular parameters, refractive outcomes, ocular aberration, optical quality, and aberration parameters, except a lower ocular trefoil in the FLACS group. In terms of these indicators, FLACS does not provide an additional clinical benefit for patients over CPS.

## 1. Introduction

Cataract extraction with intraocular lens (IOL) implantation is one of the most frequently performed surgical procedures in ophthalmology. With the increasing visual demands of cataract patients, cataract surgery has now developed from a rehabilitation surgery into a form of refractive surgery [1,2]. More and more premium IOLs have been used to reduce the patients’ dependence on glasses. The extended depth of field (EDOF) IOL, as a new type of IOL that is based on a proprietary achromatic diffraction echelette design, can extend the depth of focus instead of adding a certain focus, and is proven to be able to provide satisfactory far, intermediate, and near vision with a reduced incidence of haloes and glares [3,4].

In recent years, the clinical application of the femtosecond laser has led to new developments for cataract surgery. Combined with computer-controlled optical delivery systems, this technique has the advantages of more precise circularity and adjustability of the capsulorhexis diameter, nucleus fragmentation of lenses, arcuate keratotomy, and multi-plane self-sealing incision creation without collateral damage to surrounding tissues [5,6,7,8]. A more precise capsulotomy prior to the implantation of aspheric or multifocal IOLs contributes to better visual results, as the tilt or decentration of the IOL can led to higher asymmetrical aberrations [9,10,11,12,13,14]. Although many previous studies have been conducted to compare the safety and efficacy of femtosecond laser-assisted cataract surgery (FLACS) with conventional phacoemulsification surgery (CPS), it remains unclear whether or not the application of a femtosecond laser has an advantage for such patients with EDOF IOL implantation.

The effect and satisfaction of IOL implantation in continuous vision range are closely related to visual outcomes, as well as visual quality. The optical quality refers to the evaluation of the optical beam from the cornea to the retina [15]. It is a subjective entity and can currently be described indirectly by objective metrics such as wavefront error measurements, and visual quality metrics or functional data such as visual acuity and contrast sensitivity [16,17,18]. Wavefront analysis isolates the effect of lower-order aberrations (defocus, astigmatism) and higher-order aberrations, as well as the contribution of individual aberrations to optical quality. The objective scattering index (OSI) reflects the transparency and smoothness of the refractive media of the whole eye, along with the Strehl ratio and modulation transfer function (MTF), which are parameters of the quality of an optical system, including the human eye [19].

Therefore, the purpose of this study was to evaluate and compare the postoperative visual and refractive outcomes, wavefront aberrations, and optical quality results between FLACS and CPS in patients with a Symfony EDOF IOL implantation.

## 2. Materials and Methods

### 2.1. Patients

This prospective, consecutive cohort study was conducted at the Eye Hospital of Wenzhou Medical University (Hangzhou Branch) from October 2018 to September 2019. Thorough examinations were performed preoperatively to select eligible participants, including uncorrected distance visual acuity (UDVA), best-corrected visual acuity (BCVA), slit lamp examination, corneal tomography (Pentacam, Oculus, Germany), and optical biometry (IOL Master 700, Carl Zeiss Meditec AG, Oberkochen, Gremany). Inclusion criteria were an age over 45 years old and no active ocular disease except cataracts. Exclusion criteria were as follows: (1) patients with a history of corneal disease, retinal disease, glaucoma, and uveitis with clinically significant abnormalities; (2) patients with a history of other ophthalmic operations or combined operations (including refractive corneal surgery); (3) additional surgical procedures (including astigmatic corneal relaxing incision) during cataract surgery; (4) complications during or after cataract surgery, such as posterior capsular rupture, IOL dislocation, and posterior capsule opacification; (5) BCVA at one month postoperatively less than 20/40; (6) coexisting severe systemic diseases; and (7) known sensitivity to surgical medications used during the perioperative period. Patients who met the above criteria were given detailed information about the nature and possible consequences of this study and the characteristics of the IOL before the operation. Patients were consecutively enrolled after signing the written informed consent form, with the exception of dropout replacements. All patients were given the option to choose FLACS or CPS with the implantation of an EDOF IOL. This study was conducted adhering to the tenets of the Helsinki Declaration and was approved by the Ethics Committee of the Eye Hospital of Wenzhou Medical University (Hangzhou Branch), Wenzhou, China.

### 2.2. Intraocular Lenses

The TECNIS Symfony IOL (model ZXR00, Johnson & Johnson Vision, Santa Ana, CA, USA) is an extended depth of field IOL. It is a single-piece, UV-filtering, C-loop haptic acrylic hydrophobic folding IOL with an overall diameter of 13.0 mm and an optic diameter of 6.0 mm [20]. It can extend the depth of the focal area instead of adding focus, and it also uses achromatic technology, leading to an improvement in optical quality and at the same time correcting spherical aberrations [21,22,23]. The Haigis, SRK/T and Barrett universal II formulas were used to calculate the IOL power. By zeroing the target diopter, the optimized correlation constant of each formula was obtained. The target diopter was mild myopia.

### 2.3. Surgical Technique

All surgical procedures were carried out by the same experienced surgeon (Y.E.Z), using the standard equipment and protocol. Before the surgery, topical 0.5% levofloxacin was given four times daily to all the patients for a day. Pupil dilation was achieved with the instillation of tropicamide every 15 min three times before surgery. Topical 0.5% proparacaine hydrochloride was administered three times prior to cataract surgery.

In the CPS group, surgery was performed through a 2.2 mm clear corneal incision at 120 degrees of the corneal limbus. Capsule forceps were used to create a 5.0 to 5.5 mm continuous curvilinear capsulorhexis. In the FLACS group, patients were placed in the supine position beneath the femtosecond laser system (LenSx; Alcon Laboratories, Inc. NY). The suction ring and applanation cone were placed, and the treatment was initiated. After the laser application, the corneal incision was made manually as in the CPS group. A 5.2 mm capsulotomy centered on the dilated pupil was completed using energy of 7 µJ, and nuclear prefragmentation was performed to obtain 6 pieces in a cross pattern.

In both groups, phacoemulsification using a stand stop-and-chop technique and aspiration of the residual cortex were performed by the Infiniti apparatus (Alcon Laboratories, Inc., Fort Worth, TX, USA) and the ZXR00 IOL was implanted in the capsular bag in both groups. The residual viscoelastic material was removed from the anterior chamber and capsular bag by irrigation/aspiration. All incisions were hydrated and remained sutureless after they were checked for leakage. All the patients received a standard regimen consisting of topical antibiotics for two weeks and topical corticosteroids for one month postoperatively.

### 2.4. Patient Examinations

Visual acuity, refractive state, objective optical quality, and ocular aberration for all eyes were examined at one month postoperatively. Each examination was performed by the same technicians who performed certain tests but were blinded to the surgical approach for all operated eyes. Monocular UDVA under photopic conditions at far (5 m), intermediate (80 cm), and near distance (40 cm) were evaluated. Visual acuities were performed using Snellen eye charts with and without distance correction and transformed to the logarithm of the minimum angle of resolution (logMAR) equivalents to facilitate the statistical analysis. The difference between the spherical equivalent value after operation and the predicted refraction before operation was analyzed, named the mean error (ME) and mean absolute error (MAE). MAEs and the proportion of cases with an absolute error <2.0 D, <1.0 D, <0.5 D, and <0.25 D in each group were calculated, respectively. A wavefront aberrometry scan was performed with a ray-tracing aberrometer (iTrace, Tracey Technologies, Houston, TX, USA) at a pupil size of 4.0 mm or more under mesopic conditions. Data were recalculated to reflect a 4.0 mm pupil. The root mean square (RMS) total aberration was calculated, along with RMS total higher-order aberration, using coma, trefoil, and spherical aberration measurements [20]. Objective optical quality was assessed using a double-pass device (OQAS II, Optical Quality Analysis System II, Visiometrics SL, Terrassa, Spain), which calculated the point spread function by detecting laser light reflected from the retinal surface through a double-pass system. OQAS was used to measure the objective scattering index (OSI), modulation transfer function (MTF), Strehl ratio, and visual acuities (VAs) at 100%, 20%, and 10% contrast in both groups [21,22].

### 2.5. Statistical Analyses

Descriptive data were expressed as the mean ± standard deviation (SD). The normality of data distribution was assessed using the Shapiro–Wilk *W* test. Statistical significance was evaluated by unpaired t-test for normally distributed data and Mann–Whitney U-test for data that did not conform to a normal distribution. All the statistical analyses were performed using SPSS.23 (SPSS, IBM Corp., Chicago, IL, USA). A *p* value of <0.05 was considered as statistically significant.

## 3. Results

A total of 92 eyes of 64 patients were enrolled in this study; 35 eyes of 26 patients were treated by FLACS (the FLACS group) and 57 eyes of 38 patients were treated by CPS (the CPS group). Demographic and ophthalmological baseline characteristics of the two groups are shown in Table 1. There was no significant difference between the two groups in terms of age, sex distribution, axial length, preoperative keratometry, and anterior chamber depth (all *p* > 0.05). There was also no significant difference in the power of IOL between the two groups (*p* = 0.835).

Table 2 shows the results of UDVA at three different distances, best spectacle-corrected distance visual acuity (CDVA), and refractive results at one month postoperatively. The mean CDVA was similar between the two groups (*p* = 0.509). The mean UDVA at far and intermediate distance was around one or two letters better in the FLACS group, while the mean UDVA at near distance was two letters better in the CPS group; the histograms showed similar trends (Figure 1), but the difference was not statistically significant (all *p* > 0.05). Moreover, the mean spherical equivalent refraction after surgery was also not statistically significantly different between the two groups at one month postoperatively (*p* = 0.602).

The refractive error is the difference between the refraction predicted by the IOL formula and the actual postoperative refraction. By zeroing the target diopter, the optimized correlation constant of each formula is obtained. In the Haigis formula, optimized constants are as follows: a0 = −1.512, a1 = −0.4, a2 = 0.1. In the SRK/T formula, the optimized A constant is 118.83. Table 3 shows the ME and MAE of the two groups using the Haigis, SRK/T, and Barrett universal II formulas, respectively, and no significant difference was noted between the FLACS and CPS groups (all *p* > 0.05). The stacked histogram also showed a similar trend (Figure 2).

The incidence of postoperative ocular and internal aberrations, including total HOAs, coma, trefoil, and spherical aberration, is shown in Table 4. Assessments of ocular aberrations showed a lower ocular trefoil in the FLACS group than that in the CPS group (0.18 ± 0.09 vs. 0.27 ± 0.22, *p* = 0.033). Total HOAs, total coma, and total spherical aberrations of ocular aberrations tended to be lower in the FLACS group than those in the CPS group, but the differences were not statistically significant (all *p* > 0.05). For internal aberrations, there were still no statistically significant differences between the two groups in terms of total HOAs, total coma, and total spherical (all *p* > 0.05) (Figure 3).

The results of OQAS at one month after surgery are reported in Table 5. No significant difference was observed between the two groups in MTF, OSI, Strehl ratio, and visual acuities at 100%, 20%, and 9% contrast (all *p* > 0.05).

## 4. Discussion

Briefly, the current study suggested that there was no significant difference between the FLACS group and the PCS group in postoperative ocular parameters and refractive outcomes. We only found a lower ocular trefoil in the FLACS group, with no significant difference between the two groups considering other aberration parameters, whether ocular or internal. Our results show that FLACS does not provide an additional clinical benefit for patients over CPS.

The complex optical design of the EDOF IOL showed higher sensitivity to IOL tilt and decentration, which might lead to larger HOAs, diminishing the effect and satisfaction of EDOF IOL implantation [23,24]. FLACS has been reported to achieve unparalleled accuracy and precision in the size, shape, and position of capsulorhexis, resulting in higher refractive predictability, less corneal edema, and less IOL tilt or decentration [5,6,7,8]. To our knowledge, no previous study has comprehensively compared the clinical outcomes between FLACS and CPS with the implantation of an EDOF IOL.

In our study, clinical outcomes at one month postoperatively, including visual acuity, refractive result, objective optical quality, and ocular aberration, were evaluated and compared between the FLACS group and the CPS group. The results showed that UDVA, CDVA, and refractive outcomes had no statistical difference between the two groups. Meanwhile, the mean uncorrected vision for intermediate and far distance was around one or two letters better in the FLACS group and the mean UDVA at near distance was two letters better in the CPS group, but the difference was not statistically significant. Table 3 shows the ME and MAE of the two groups using the Haigis, SRK/T, and Barrett universal II formulas, respectively, and no significant difference was noted. These results are consistent with those of previous studies [6,13,23,24]. A multi-center participant-masked, randomized superiority study with 907 patients (1476 eyes) showed no significant differences in visual, refractive, and corneal astigmatism change or anatomical complications between patients treated by FLACS and patients treated by CPS [25]. However, some studies suggested that femtosecond laser treatments may be associated with a lower absolute spherical equivalent refraction error [13,26]. The difference in results may be due to differences in the proficiency of the surgeon, laser platform used, and the different IOL used. Conrad Hengerer et al. used the monofocal spherical IOL, and Espaillat et al. used toric monofocal IOLs and multifocal IOLs, while we used the Symfony EDOF IOL [13,27]. Since all eyes included in our study were subjected to the same IOL, the optical properties of the IOL were unlikely to bias the comparison between the two groups.

Wavefront aberration analysis isolates the effects of lower-order aberrations (defocus and astigmatism) and HOAs. Although HOAs comprise several components, the overall effect on visual performance is the combination of aberrations in the ocular or internal optics [28]. Larger aberrations may induce refractive visual defects. Previous studies have used the iTrace to measure aberrations of diffractive multifocal IOLs [29,30]. The iTrace is a ray-tracing type of aberrometer, which projects near-infrared laser beams into the eye. The laser beams reflected from the retina are measured by a position-sensitive detector. Using the reflected beam, the aberrometer calculates forward aberrations and, even if the beams scatter as they pass the diffractive multifocal IOL, the retinal image is less affected [20]. In the current study, we found a lower ocular trefoil in the FLACS group and no significant difference between the two groups considering other aberration parameters, whether ocular or internal. Although all ocular parameters were lower in the femtosecond group, we only found a significant improvement in ocular trefoil. Moreover, no significant difference was found in the total HOAs, which is consistent with the previous studies [19,23]. Theoretically, laser capsulotomy could improve IOL centration or prevent tilt within the capsular bag by providing a complete overlap of the IOL edge by the anterior capsule [27,31]. Tilted or decentered IOLs have been associated with an increase in HOAs, particularly coma [13,32,33]. This theory might account for the improvement in ocular trefoil in the FLACS group. Overall, our results suggested that a femtosecond laser has some advantages, but it seems to be negligible. However, as far as we know, HOAs were reported to be lower with FLACS in two studies in the literature [13,32,33]. We believe that this discrepancy may be due to the choice of different IOLs, as both studies focused on toric IOLs. Mojzis’s analysis of internal astigmatism and high-order aberrations in multifocal toric IOL supported this assumption [34].

Other metrics to quantify optical quality include the Strehl ratio and MTF, which predict how optical systems modulate a point image and contrast sensitivity [34,35]. We also compared the effect of IOL on vision quality in the two groups with optical OQAS. There was still no significant difference observed between the two groups in MTF, OSI, Strehl ratio, and visual acuities at 100%, 20%, and 9% contrast. Our results are consistent with the conclusions of Wang et al. [15]. However, the results reported by Miháltz et al. are inconsistent with ours; they compared 48 eyes treated with FLACS and 51 eyes treated with CPS and found a statistically significant increase in Strehl ratio and MTF values in the FLACS group six months after surgery, suggesting better image quality in these patients [19]. We believe that the difference is derived from surgical differences and different IOL choices, as Miháltz’s study also found that the IOL (three-piece acrylic spherical IOL) was tilted more severely in the CPS group.

Admittedly, the current study has some limitations that need to be acknowledged. First, the relatively small number of patients and short follow-up period may have diminished the power of the current study. The comparison of optical quality between the two groups needs a long-term follow-up. Second, the current results might have selection bias as the eyes were not randomized to the FLACS group or the CPS group. The relatively high cost of FLACS precluded randomization, although there were no differences in demographic or ophthalmological baseline characteristics between the two groups. Additionally, we did not ask the patients to fill out satisfaction questionnaires and we did not evaluate the patients’ subjective feelings; further research might be necessary to investigate these results.

## 5. Conclusions

Our results suggested that there was no significant difference between the FLACS group and the CPS group in postoperative ocular parameters and refractive outcomes. Moreover, we only found a lower ocular trefoil in the FLACS group, with no significant difference between the two groups considering other aberration parameters, whether ocular or internal. In terms of these indicators, there is little difference between FLACS and CPS with the implantation of an EDOF IOL. Our results show that FLACS does not provide an additional clinical benefit for patients over CPS.

## Figures and Tables

**Figure 1 jpm-13-00400-f001:**
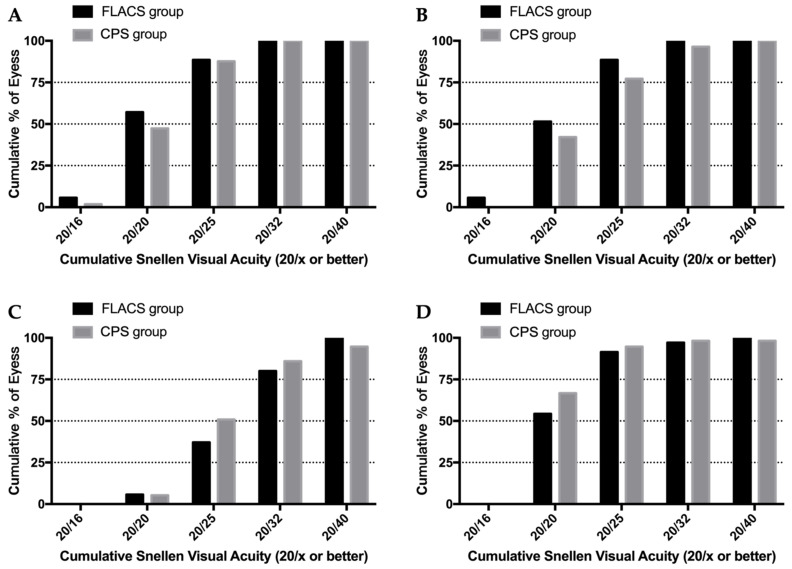
Postoperative visual acuities and refractive results. (**A**) Far vision uncorrected distance visual acuity (UDVA); (**B**) intermediate vision UDVA; (**C**) near vision UDVA; (**D**) corrected distance visual acuity (CDVA).

**Figure 2 jpm-13-00400-f002:**
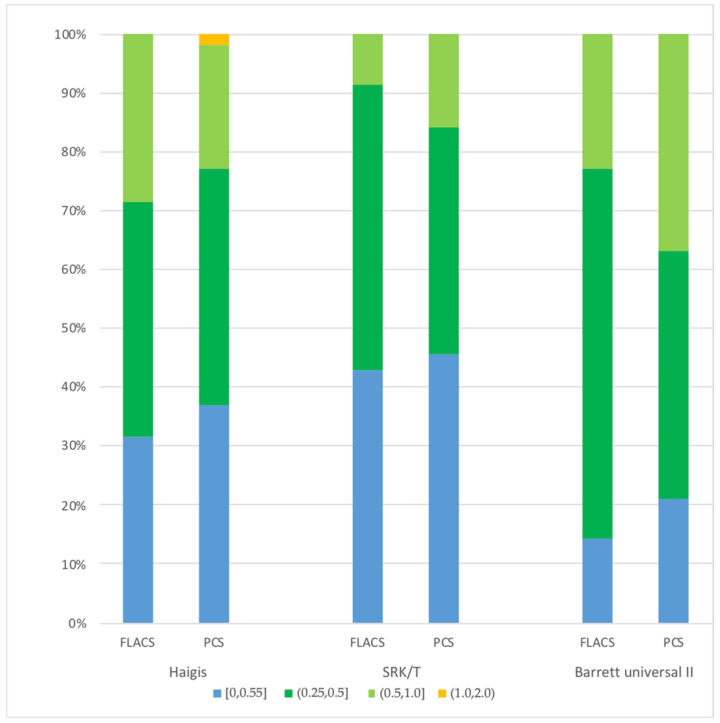
Stacked histogram comparing the percentage of cases within a given diopter range of absolute predicted spherical equivalent refraction outcome in both groups.

**Figure 3 jpm-13-00400-f003:**
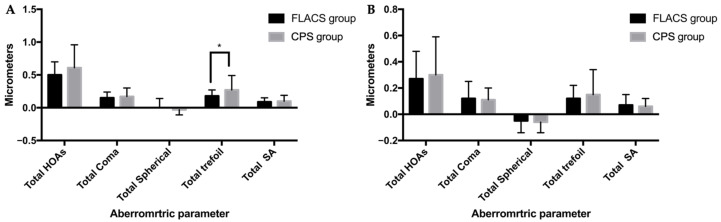
(**A**) Ocular aberrations. Asterisks indicate statistically significant differences between the two groups. (**B**) Internal aberrations. Asterisks indicate statistically significant differences between the two groups. (HOAs = higher-order aberrations; SA = spherical aberration).

**Table 1 jpm-13-00400-t001:** Patient’s demographics and ophthalmological baseline characteristics.

Characteristics	FLACS Group	CPS Group	*p*-Value
Eyes/patients (n)	35/26	57/38	-
Age (y)	65.20 ± 12.88	70.07 ± 8.35	0.051
Gender (M/F)	11/15	16/22	0.987
IOL power (D)	21.21 ± 2.94	21.10 ± 2.42	0.835
Axial length (mm)	23.46 ± 1.01	23.62 ± 1.10	0.506
Average K (D)	44.42 ± 1.14	44.15 ± 1.84	0.389
Astigmatism (D)	−0.53 ± 0.35	−0.55 ± 0.31	0.781
ACD (mm)	3.08 ± 0.40	2.99 ± 0.38	0.291

IOL: intraocular lens, ACD: anterior chamber depth.

**Table 2 jpm-13-00400-t002:** Postoperative visual acuities (logMAR) and refractive results.

Parameter	FLACS Group	CPS Group	*p*-Value
UDVA			
Far vision	0.05 ± 0.08	0.06 ± 0.08	0.256
Intermediate vision	0.06 ± 0.08	0.08 ± 0.09	0.162
Near vision	0.18 ± 0.09	0.16 ± 0.09	0.339
CDVA	0.03 ± 0.04	0.03 ± 0.07	0.509
Spherical equivalent (D)	−0.61 ± 0.57	−0.55 ± 0.51	0.602

logMAR: logarithm of the minimum angle of resolution, UDVA: uncorrected distance visual acuity, CDVA: corrected distance visual acuity.

**Table 3 jpm-13-00400-t003:** Postoperative refractive error.

	Haigis			SRK/T			Barrett Universal II		
	FLACS Group	CPS Group	*p*	FLACS Group	CPS Group	*p*	FLACS Group	CPS Group	*p*
ME	−0.31 ± 0.61	−0.23 ± 0.50	0.476	−0.38 ± 0.63	−0.29 ± 0.58	0.467	−0.41 ± 0.22	−0.42 ± 0.20	0.772
MAE	0.35 ± 0.21	0.34 ± 0.22	0.910	0.29 ± 0.16	0.30 ± 0.20	0.820	0.43 ± 0.19	0.42 ± 0.20	0.915

ME: mean error, MAE: mean absolute error.

**Table 4 jpm-13-00400-t004:** Postoperative iTrace measurement results.

Parameter	FLACS Group	CPS Group	*p*-Value
Ocular aberrations			
Total HOAs	0.50 ± 0.20	0.61 ± 0.35	0.076
Total coma	0.15 ± 0.09	0.17 ± 0.13	0.317
Total spherical	0.00 ± 0.14	−0.03 ± 0.08	0.229
Total trefoil	0.18 ± 0.09	0.27 ± 0.22	0.033 *
Total SA	0.09 ± 0.06	0.10 ± 0.09	0.443
Internal aberrations			
Total HOAs	0.27 ± 0.21	0.30 ± 0.29	0.621
Total coma	0.12 ± 0.13	0.11 ± 0.09	0.907
Total spherical	−0.05 ± 0.09	−0.06 ± 0.08	0.717

HOAs: higher-order aberrations, SA: spherical aberration, * denotes statistically significant result.

**Table 5 jpm-13-00400-t005:** Postoperative OQAS measurement results.

Parameter	FLACS Group	CPS Group	*p*-Value
MTF	35.06 ± 12.08	35.82 ± 11.46	0.764
OSI	1.32 ±0.88	1.27 ± 0.89	0.681
SR	0.18 ± 0.07	0.18 ± 0.07	0.894
VA 100%	1.21 ± 0.40	1.23 ± 0.38	0.856
VA 20%	0.90 ± 0.36	0.88 ± 0.32	0.786
VA 9%	0.50 ± 0.22	0.50 ± 0.22	0.902

MTF: modulation transfer function, OSI: objective scattering index, SR: Strehl ratio, VA 100%, 20%, 9%: visual acuities at 100%, 20%, and 9% contrast.

## Data Availability

The data are available upon reasonable request.

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
