# Peer review of "Femtosecond Laser-Assisted Cataract Surgery versus Conventional Phacoemulsification Surgery: Clinical Outcomes with EDOF IOLs"

_jpm, 2023, doi:10.3390/jpm13030400_

Round 1
Reviewer 1 Report
The manuscript is interesting. Please explain why did you not performed all the cataract surgery with the sam instrument "In both the FLACS and CPS groups, phacoemulsification using a stand stop-and- 113 chop technique (Stellaris system, Bausch & Lomb, Rochester) and aspiration of the resid- 114 ual cortex were performed by the Infiniti (Alcon Laboratories, Inc.) "
Author Response
We are very grateful for the comments made during review of our manuscript. We realized that the description of surgical technique in our manuscript was ambiguous. Actually, all phacoemulsification in the current study were completed using the Infiniti (Alcon Laboratories, Inc.). We apologise for our careless description and revised it in line 115.
Reviewer 2 Report
The relevance of the problem addressed in the article is due to the aging of the population in developed countries and the associated high incidence of senile cataracts. Cataract extraction with intraocular lens (IOL) implantation is one of the most frequently performed surgical procedures not only in ophthalmology, but also in medicine in general. This surgical intervention is considered to be safe and effective today. The design of the IOL and the position of the lens on the visual axis are taken into account by including the A-constant or anterior chamber depth, as well as a number of other biometrics, in the calculation. The result of the operation is usually expressed in increased visual acuity and/or improved quality of life after the operation. Therefore, the research topic is all relevant and timely.
But before recommending the article for publication in the Journal of Personalized Medicine, I would like the author to comment on the following: Reading abstract/conclusions, the meaning of the article becomes unclear. A comparison is made that shows no difference. But the meaning of femtosecond processing and IOL is different.
Between 77% and 90% of all eyes after IOL implantation achieve visual acuity of 0.5 or better. However, despite numerous advances in cataract surgery, ill-fitting IOL power remains one of the most common reasons for replacing them in the early and late postoperative period. Thus, modern statistical studies show that only in 72.3% of cases of cataract removal, patients after surgery have a refraction within 1.0 diopters and in 6.4% the refraction exceeds 2.0 diopters from the planned one.
Although the error rate in IOL power fitting has been declining in recent years, mismatched IOL power remains a serious problem. Causes of error in determining IOL power include the following: technological (for example, incorrect measurement of the axial length of the eye, determination of the corneal power or A-constants), mechanical (for example, placing the lens in a position other than expected before surgery, or axis deviation caused by unwanted rotation of the toric IOL), as well as the unpredictable impact of wound healing on the effective position of the lens in the capsular bag, especially capsule shrinkage, pre-existing or iatrogenically induced astigmatism without treatment, procedural or technological errors (for example, incorrect IOL placement or incorrect labeling of the IOL by the manufacturer). ) and any combination of them. Incorrect corneal measurements followed by errors in axial length measurements are particularly cited as the main culprits for misdetermining IOL power. Current laser refractive surgeries, such as photorefractive keratectomy, in situ laser keratomileusis and others, are quite common all over the world. However, it should be noted that the consequences of these operations include topographic changes in the cornea and a decrease in the accuracy of preoperative keratometric measurements in patients who are scheduled for cataract surgery.
A significant number of patients with myopia who underwent refractive surgery also determines the number of postoperative refractive surprises as these patients age and the need for surgical treatment of cataracts. The urgency of the problem is also due to the development of cataract surgery as a refractive direction due to the need to correct presbyopia in relatively young patients of working age who undergo removal of the transparent lens with IOL implantation. As a result, most IOLs today are available in 0.5 diopter increments and tend to overcorrect or undercorrect patients. While this provides a suitable arsenal to provide patients with emmetropia today, it may not be acceptable in the future when patients seek spectacle independence. With this in mind, it can be argued that the future of cataract and refractive surgery should include the use of postoperatively adjustable IOLs.
The basic principle behind this new technology is that the power of the patient's IOL can be adjusted post-implantation using new invasive tools to provide near-perfect vision. This unique technology, theoretically, will reduce the amount of refractive "surprises" after cataract surgery.
In connection with the foregoing, it is proposed to modify the Introduction, taking into account the methods of non-surgical correction of the optical power of the IOL, based on creating the required profile of the refractive index of the lens by applying microstructures to it with a femtosecond laser. As shown by Bille, Josef F et al. the deposition of structures in the volume of the IOL leads to a change in the hydrophilicity of the target area, which makes it possible to change the refractive index. This effect, combined with the applied two-dimensional pattern, allows the creation of a refractive or diffractive lens within the material. This method will allow noninvasively, by quickly scanning an intraocular lens placed directly in the patient's eye, to correct its optical power.
Author Response
We are very grateful for the comments made during review of our manuscript.
As the reviewer mentioned above, cataract surgery is one of the most common ophthalmic procedures in clinical practice. Postoperative refractive outcomes remain a key concern for surgeons. Improvements in surgical technique, new IOL technologies, enhanced biometric methods, and advanced methods of IOL power calculation have led to modern cataract surgery being a refined procedure. There is an increasing patient expectation of excellent postoperative outcomes and high demand for spectacle independence. However, unsatisfactory visual outcomes due to residual refractive errors may occur and correction of large refractive errors requires additional procedures. Various pre-, intra-, and postoperative factors influence refractive outcomes after cataract surgery. Although the error rate in IOL power fitting has been declining in recent years, mismatched IOL power remains a serious problem.
There are some new IOL technologies available or under development, which aim to mitigate the problem of incorrect IOL power and produce safe and efficacious refractive results, including IOL technologies that can be adjusted using secondary surgical procedures, such as the multicomponent IOL, the mechanically adjustable IOL, and the repeatedly adjustable IOL; IOLs that can be adjusted noninvasively in the postoperative setting, such as the magnetically adjustable IOL, the liquid crystal IOLs with wireless control; and IOLs that can be adjusted using the femtosecond laser or 2-photon chemistry. The EDOF IOL TECNIS Symfony (Johnson & Johnson), based on a proprietary achromatic diffraction echelette design, is proven to be capable of restoring the patient’s far, intermediate, and part of near vision. The latest-generation (2.0) Light-Adjustable Lens (RxSight) was recently introduced into clinical practice, with the first results being very encouraging. We anticipate those new IOL technologies may become the mainstay of cataract treatment in the years to come.
References:
- Dick HB, Gerste RD. Future Intraocular Lens Technologies. Ophthalmology. 2021 Nov;128(11):e206-e213.
- Villegas EA, Alcon E, Rubio E, Marín JM, Artal P. Refractive accuracy with light-adjustable intraocular lenses. J Cataract Refract Surg. 2014 Jul;40(7):1075-84.e2.
- Lee NS, Ong K. Factors contributing to long-term refractive error after cataract surgery. Int Ophthalmol. 2023 Jan 2.
- Khoramnia R, Auffarth G, Łabuz G, Pettit G, Suryakumar R. Refractive Outcomes after Cataract Surgery. Diagnostics (Basel). 2022 Jan 19;12(2):243.
Round 2
Reviewer 2 Report
Good luck.